# Altered Development of Gut Microbiota and Gastrointestinal Inflammation in Children with Post-Operative Hirschsprung’s Disease

**DOI:** 10.3390/ijms262110570

**Published:** 2025-10-30

**Authors:** Caitlin E. Murphy, Michael J. Coffey, Quinlan Chen, Susan Adams, Chee Y. Ooi, Josie van Dorst

**Affiliations:** 1School of Clinical Medicine, Discipline of Paediatrics and Child Health, UNSW Medicine and Health, University of New South Wales, Randwick 2052, Australiamichael.coffey@health.nsw.gov.au (M.J.C.); susan.adams@unsw.edu.au (S.A.); keith.ooi@unsw.edu.au (C.Y.O.); 2Department of Gastroenterology, Sydney Children’s Hospital, Randwick 2031, Australia; 3Department of Paediatric Surgery, Sydney Children’s Hospital, Randwick 2031, Australia

**Keywords:** Hirschsprung’s disease, microbiome, inflammation, diet, paediatric, functional disease, gastrointestinal disease

## Abstract

Gastrointestinal symptoms often persist in children with Hirschsprung’s disease (HD) even after “corrective” pull-through surgery. Alteration of the gut microbiota (“dysbiosis”) has emerged as a potential contributing factor. Animal studies show gut ecosystem changes that are both intrinsic to HD and caused by bowel resection itself, but human studies comparing the intestinal microbiota of children with HD and healthy children are limited. We collected food frequency dietary surveys, clinical and symptom data, and stool samples from 15 post-operative children with HD and 15 healthy controls (HCs). We performed 16S rRNA gene sequencing from the stool samples and quantified faecal calprotectin as a measure of gastrointestinal inflammation. Despite no global changes in the microbiota between HD and HC cohorts and no differences between individuals with and without a history of HD-associated enterocolitis (HAEC), we identified evidence of altered microbiota development and inflammatory trajectories in HD. In HCs, alpha diversity increased with age (r = 0.83, *p* < 0.001), while calprotectin levels declined (Spearman’s ρ = −0.53, *p* = 0.04). These age-related patterns were absent in HD. Across the combined cohort, lower alpha diversity was associated with higher faecal calprotectin (Spearman’s ρ = −0.47, *p* = 0.01). In HD, *Fusobacteria* abundance showed a strong positive correlation with calprotectin (Spearman’s ρ = 0.76, adjusted *p* = 0.02). Pediatric Quality of Life (PedsQL) and gastrointestinal disease-specific symptom scores were lower in HD compared to HC but were not directly linked to microbial diversity or inflammation. Overall, we observed a divergence from healthy peers in the typical developmental trajectory of gut microbial communities and inflammation in children with HD that may involve *Fusobacteria*. Children with HD reported reduced health-related quality of life compared with HC, consistent with ongoing gastrointestinal symptoms. No microbiota differences were associated with HAEC history, though this may reflect limited sample size.

## 1. Introduction

Hirschsprung’s disease (HD) is a congenital condition characterised by a lack of parasympathetic innervation in part, or all of the colon, thereby restricting peristalsis and the normal passage of stool. In almost all cases, the first indication of HD is a failure to pass meconium in newborns, leading to functional bowel obstruction that can be life-threatening [1]. Untreated, all infants and children with HD have a significantly increased risk of bowel perforation, enterocolitis and severe infection [2,3]. An operation to remove the affected bowel and “pull-through” ganglionated bowel to just above the dentate line is required to mitigate this risk. Pull-through surgery has been shown to improve long-term functional outcomes for patients [4,5,6]. However, gastrointestinal symptoms are likely to persist even after the operation, resulting in reduced quality of life [4,6,7,8].

It has been suggested that the disrupted neuroimmune development and regulation, compromised mucosal barrier function and extended transit time of stool through the large intestine disrupt the microbial milieu in HD, exacerbating gastrointestinal (GI) symptoms and inflammation caused by the bowel obstruction itself [9]. Findings from animal and limited human studies support the occurrence of gut dysbiosis in HD and suggest that HD-related changes to the gut microbiota are not fully remedied by or are potentially worsened by pull-through surgery. Specifically, increases in *Bacteroidetes* and *Proteobacteria* (including *Prevotella*, *Escherichia* and *Pseudomonas*) and decreases in *Firmicutes* (including the beneficial anti-inflammatory *Lactobacillus* genus) have been reported in post-operative HD [10,11], as well as in animal models of HD [12,13,14], and in humans with non-HD-related bowel obstructions [15]. Together, the evidence suggests changes to the microbiota in HD likely occur prior to surgery, persist after surgery, and contribute to ongoing bowel dysfunction. Critically, gut dysbiosis in children with HD may predispose them to developing HD-associated enterocolitis (HAEC) [16,17], a potentially life-threatening inflammatory condition of the colon.

In this study, we sought to (1) compare microbial diversity and composition in the gut between post-operative children with HD and healthy controls, (2) identify potential associations between diet, GI symptoms, inflammation and gut microbiota in HD and (3) compare the microbiota profiles of children with HD with and without a history of recurrent HAEC. We hypothesise that microbiota profiles will have significant associations with HD and HD-related inflammation.

## 2. Results

### 2.1. Demographics

In line with HD epidemiology, more males with HD were recruited than females 13/15 (87%). Of the 15 children recruited, 11 (73%) had short segment length affected and the average number of years (SD) since pull-through surgery was 4.7 (4.1). Eight children (53%) had a history of HAEC. The 15 children recruited were age- and gender-matched with 15 healthy controls (HCs). There were no significant differences between HD and HC anthropometrics. A summary of cohort demographics is presented in Table 1.

### 2.2. Alpha Diversity of the Gut Microbiota Is Unchanged but Does Not Follow the Typical Age-Related Trajectory in Children with HD

Alpha diversity indices were not significantly different between HD and HC children (richness: mean difference −15.0 [95% confidence interval: −97.32–67.38], *p* = 0.71; Shannon–Weaver: mean difference −0.16 [−0.57–0.90], *p* = 0.65) (Figure 1A,B). Alpha diversity increased with age across our HC cohort (richness: *r* = 0.79, *p* < 0.004; Shannon–Weaver: *r* = 0.69, *p* < 0.004) but not in HD (richness: *r* = 0.19, *p* = 0.51; Shannon–Weaver: *r* = 0.03, *p* = 0.91) (richness: Fisher’s z = 2.21, *p* = 0.03; Shannon–Weaver: Fisher’s z = 2.05, *p* = 0.04) (Figure 1C,D). Alpha diversity was not significantly associated with gender (richness: *r* = 0.89; Shannon–Weaver: *p* = 0.57), recent antibiotics (*r* = 0.03, *p* = 0.9) or PedsQL symptom scores (richness: *r* = −0.29, *p* = 0.42; Shannon–Weaver: *r* = −0.26, *p* = 0.48).

### 2.3. Bacterial Composition of the Gut Microbiota in Paediatric HD Is Unchanged at All Taxonomic Levels

Differences in beta diversity between HD and HCs were not detected in our cohort (Bray–Curtis dissimilarity (relative abundance): *F* = 0.80, *p* = 0.75; Binary Bray–Curtis dissimilarity (presence/absence): *F* = 0.66, *p* = 0.88) (Figure 2). Additionally, we found no association between beta diversity and age (Bray–Curtis: *F* = 1.24, *p* = 0.18; Binary Bray–Curtis: *F* = 1.03, *p* = 0.35) or gender (Bray–Curtis: *F* = 1.12, *p* = 0.24; Binary Bray–Curtis: *F* = 1.75, *p* = 0.06). The most abundant bacterial phyla and genera in our cohort are listed in Appendix A. Briefly, *Bacteroidetes*, *Firmicutes* and *Actinobacteria* phyla dominated the majority of stool samples while *Bacteroides*, *Rhodoccocus* and *Faecalibacterium* were the most abundant genera overall. Notably, the stool sample from one HD subject (HD13) showed little diversity and was largely composed of the *Rhodoccocus* genus from the *Actinobacteria* phylum. Removing this sample for leave-one-out sensitivity analysis did not alter the results of statistical comparisons. Thus, the sample was included in analysis. ANCOM analysis confirmed no differences in bacterial abundance at any level (phylum, class, order, family or genus) (all q > 0.05).

### 2.4. PedsQL Was Reduced in HD Compared to HC

The HD group scored significantly lower on the PedsQL, indicating higher prevalence and severity of GI symptoms (*p* = 0.005) and reduced quality of life (*p* < 0.02) (Table 2). PedsQL symptom scores in HD were not associated with Calprotectin (*r* = 0.56, *p* = 0.09) or years since surgery (*r* = 0.02, *p* = 0.9).

### 2.5. Children with HD Show No Overall Increase in the Gastrointestinal Inflammatory Marker Calprotectin

Two HCs and four children with HD had clinically elevated faecal calprotectin (>160 μg/g) as per the specified clinical thresholds (*p* = 0.65). Children with HD did not differ from HCs in calprotectin levels overall (Wilcoxon rank-sum test, W = 129, *p* = 0.068) (Wilcoxon rank-sum test, W = 129, *p* = 0.07) (Figure 3A). Given that children under 4 have higher and more variable levels of faecal calprotectin [18], we also compared levels in children under 4 and over 4 separately but found no diagnostic difference in either age group (under 4 yrs: V = 68, *p* > 0.99; over 4 yrs: V = 62, *p* = 0.26). However, higher levels of calprotectin in our cohort were associated with lower levels of alpha diversity even after accounting for age (richness: Spearman’s *ρ* = −0.39, *p* = 0.03; Shannon–Weaver: Spearman’s *ρ* = −0.33, *p* < 0.04). In analysing putative relationships between GI inflammation and specific members of the microbiota, we found that calprotectin was inversely correlated with abundance of the *Oscillibacter* genus in HCs (Spearman’s *ρ* = −0.84, q = 0.03) and positively correlated with abundance of the *Fusobacteria* phylum in HD (Spearman’s *ρ* = 0.76, q = 0.02). An inverse relationship between age and calprotectin was observed in controls only (Spearman’s *ρ* = −0.56, *p* = 0.035; Figure 3B), and calprotectin was not associated with height or weight in either group (all *p* > 0.11).

### 2.6. History of Recurrent HAEC Is Not Associated with Changes to the Microbiota or with Gastrointestinal Inflammation

Of the twelve HD children with available clinical data, eight had a history of recurrent HAEC. This group showed no difference in alpha diversity (richness: t = 1.92, *p* = 0.09; Shannon–Weaver: t = 1.37, *p* = 0.20) or beta diversity (Bray–Curtis: F = 0.03, *p* = 0.88; Binary Bray–Curtis: F < 0.01, *p* = 0.96) compared to HD children without a history of HAEC. ANCOM analyses confirmed no significant differences in differential abundance of bacteria at any taxonomic level. History of recurrent HAEC was not associated with changes in faecal calprotectin (t = −1.04, *p* = 0.32).

### 2.7. Dietary Intake Does Not Differ Between Children with HD and Controls and Is Not Associated with Gastrointestinal Inflammation or Bacterial Abundance

We found no difference in total dietary intake, macro- or micro-nutrient intake between HD and HCs (Figure 4, Appendix A). There were also no significant associations between dietary variables and faecal calprotectin (q > 0.42) or between dietary variables and count data for specific phyla or genera (q > 0.07).

## 3. Discussion

Gut microbiota imbalances are proposed to contribute to persistent GI symptoms in post-operative children with HD. We found that unlike HC, children with HD do not acquire increasing gut microbiota diversity with age. This disruption in microbial maturation is paralleled by faecal calprotectin that persists in HD, as opposed to the expected decline observed in HC, with faecal calprotectin concentrations negatively correlated to alpha diversity. Importantly, children with HD reported lower quality of life, especially in domains linked to gastrointestinal symptoms.

Animal models of HD have reported reduced microbiota diversity following surgery and in HAEC-prone mice, but clinical evidence in children remains scarce and inconsistent [9]. Small patient cohorts, variable study designs and different specimen types complicate comparisons across studies [10,16,17]. Our findings demonstrate that children with HD show a disrupted developmental trajectory of diversity, rather than a static difference at a single time point.

Although we did not observe broad compositional shifts, exploratory analyses identified taxa of interest. *Fusobacteria* correlated positively with calprotectin, suggesting a pro-inflammatory role, consistent with reported associations with IBD and pro-inflammatory pathways [19,20,21]. *Oscillibacter* showed inverse associations, consistent with protective, anti-inflammatory effects reported elsewhere [22,23]. While both taxa are relevant to gastrointestinal health, underscoring their potential importance in HD pathophysiology, larger studies are needed to assess causality.

We did not identify stool microbiota signatures predictive of recurrent HAEC, contrasting with tissue-based studies in younger patients at the time of surgery [16,17]. Differences in biospecimen type [19], time of sampling and patient age likely explain these discrepancies. Our single stool collection time point was outside of active HAEC, as opposed to stool samples obtained at the time of emergency or routine surgery in [16,17]. Likewise, the average age of participants in our study was 5.6 years (SD 5.0) compared to <12 months in [16,17].

Notably, this study is the first to examine microbiota alongside quality of life in HD, highlighting persistent symptom burden despite limited evidence of compositional change. Our results are in line with a recent systematic review highlighting impaired physical functioning and reduced GI disease-specific PedsQL scores in HD, with bowel dysfunction identified as the strongest determinant of reduced quality of life [24].

This study was limited by its small sample size, cross-sectional design, and reliance on stool sampling, which may miss mucosa-associated alterations. The combination of low participant numbers and high inter-individual variability increases the risk of false negatives (type II errors). Nonetheless, careful control for confounding variables including diet, antibiotic exposure, age and gender strengthens confidence in the findings. Importantly, HD is rare and comparable studies in the field have included similar or smaller cohorts, underscoring the challenges of research in this population. All eligible participants meeting the inclusion and exclusion criteria were invited to participate to maximise the sample size.

## 4. Materials and Methods

### 4.1. Study Design

We conducted a cross-sectional, controlled observational study, comparing children with HD to an age- and sex-matched cohort of healthy controls (HCs) to minimise demographic confounding. The study formed part of the EARTH (Evaluating the Alimentary and Respiratory Tracts in Health and Disease) programme (ethics approval HREC/18/SCHN/26, approved on 14 March 2018) described previously [25] and was conducted at Sydney Children’s Hospital in Randwick in 2019. Inclusion criteria included the following: (i) aged between 0 and 18 years; (ii) diagnosed with HD based on rectal biopsy and post-pull-through surgery (HD group) or free of any chronic health condition (HC group) and (iii) provided informed consent if 16 years of age or older or a parent(s)/carer(s) provided informed consent on behalf of children under 16. Exclusion criteria included the following: (i) children with more than one concurrent or unrelated chronic disease; (ii) inability to comply with study requirements and (iii) participant/guardian unable to speak English or have an English reading level lower than 12 years of age equivalency.

### 4.2. Sample Collection and Analysis

All subjects were requested to provide a stool sample, collected as per the EARTH programme protocol [25]. Samples were thawed and homogenised prior to DNA extraction using the QIAamp Fast DNA Stool Mini Kit (QIAGEN, Hilden, Germany) as per manufacturer instructions. 16S ribosomal RNA gene sequencing and amplification was performed using primers 515 F (GTGYCAGCMGCCGCGGTAA) and (GTGYCAGCMGCCGCGGTAA) and 806 R (GGACTACNVGGGTWTCTAAT) spanning the V4 region, on the Illumina MiSeq platform (Illumin, San Diago, CA, USA) at the Ramaciotti Centre for Genomics (University of New South Wales, Sydney, Australia). Quality filtering was performed according to the thresholds outlined in [25,26]. Processed sequences were clustered in unique sequences (zero-distance operational taxonomic unit zOTU) with the unoise2 algorithm implemented in USEARCH v9.2.64. After chimaera removal, sequences were then classified by BLASTn alignment against the SILVA database. Concatenated sequences of all sequences were mapped on the final set of zOTUs to calculate the abundance of each zOTU for each sample. Calprotectin, a protein marker of GI inflammation was measured from stool samples using a monoclonal enzyme-linked immunosorbent assay (EK-CAL Calprotectin ELISA, Bühlmann, Switzerland) according to manufacturer instructions.

### 4.3. Clinical Surveys and Analysis

Each participant or their parent was asked to complete a clinical questionnaire for the collection of demographic and anthropometric information. The questionnaire was designed and distributed through Qualtrics (Qualtrics, Provo, UT, USA). Included in the questionnaire was the Pediatric Quality of Life Inventory (PedsQL; Infant Physical Symptoms or Gastrointestinal Symptoms Module) to quantify severity of GI symptoms and their effect on quality of life [27,28]. Scores were calculated out of 100, where 100 indicated low presence or severity of symptoms and high quality of life. To assess the effect of diet on our variables of interest, dietary intakes were quantified by the Australian Child and Adolescent Eating Survey (ACAES) for children over 2 years of age, and by dietitian-administered 24 h food recall for children under 2, as per the study in [29]. Clinical data for HD patients was obtained through medical chart review (clinician confirmed) and included affected colonic segment length, syndromic status and history of HAEC. HAEC was defined clinically and based on discharge diagnosis: characteristic symptoms of fever, lethargy, vomiting and diarrhoea, often with blood, and in the absence of infectious cause (negative stool microbiology).

### 4.4. Statistical Methods

Statistical analyses were performed in RStudio (v3.6.0). Categorical variables were compared using Fisher’s Exact Test for count data; continuous variables were evaluated with the Shapiro–Wilk test of normality and analysed according to distribution with Student’s *t*-test or the Mann–Whitney U test for parametric and nonparametric data, respectively. A *p*-value of <0.05 was considered statistically significant. To account for any residual variation not fully addressed by matching, differences between groups were assessed using generalised linear models with age (continuous) included as a covariate. Alpha diversity, a measure of the number of unique species, was assessed in stool via richness (number of zOTUs) and the Shannon–Weaver index. Beta diversity, a measure of composition and variety of microbes in the gut, was calculated using Bray–Curtis dissimilarity to generate nonmetric multidimensional scaling (NMDS) plots. Permutational multivariate analysis of variance (PERMANOVA) tests were conducted using the adonis2 function (vegan package) to determine differences in beta diversity between HD and HC cohorts. Differential abundances of taxonomy between cohorts were determined using the analysis of composition of microbiomes (ANCOM) package v1.1–3 and corrected for multiple testing (false discovery rate (FDR) q < 0.05). Correlations between two continuous variables were performed using Spearman correlations (adjusted *p*-values were produced using the Benjamini–Hochberg correction for multiple testing q < 0.05). Fisher’s r-to-z transformation was used to assess the significance of the difference between correlations in HD and HCs [30]. Graphs were produced using the ggplot2 package (v3.5.1).

## 5. Conclusions

Children with HD diverge from healthy peers in the typical developmental trajectory of gut microbial diversity and inflammation, potentially mediated by *Fusobacteria*. They also exhibit lower PedsQL and gastrointestinal symptom scores than HC, though these were not directly linked to microbial diversity or inflammation. While these findings are constrained by sample size, they highlight the need for larger, longitudinal and mechanistic studies integrating microbiota, inflammatory and clinical outcomes to define causal pathways and identify therapeutic strategies to improve gastrointestinal health and quality of life in children with HD.

## Figures and Tables

**Figure 1 ijms-26-10570-f001:**
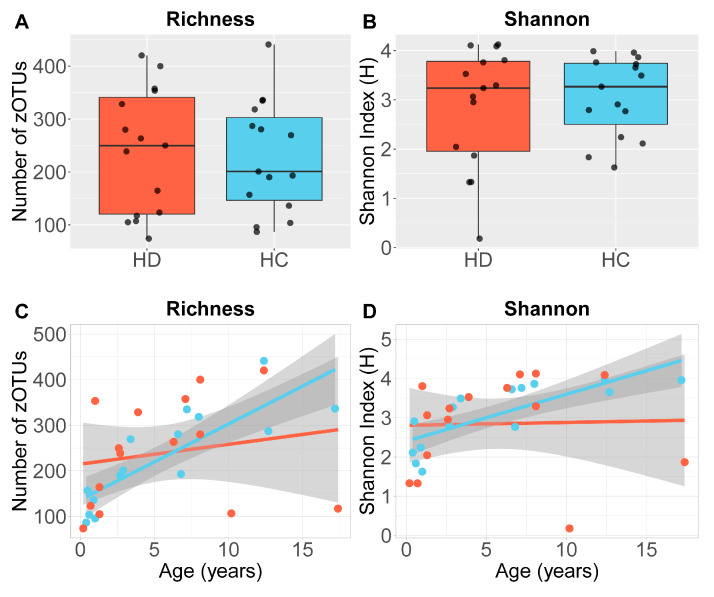
Alpha diversity in stool in Hirschsprung’s disease (HD) and healthy controls (HCs). (**A**) Richness and (**B**) Shannon–Weaver index were not different between groups. Boxplots depict the middle 50% of values, with group medians denoted by horizontal lines. Age was positively correlated with both (**C**) Species Richness and (**D**) Shannon–Weaver index in the HC group, but not in the HD group. Solid lines represent generalised linear regression model-predicted values for HD (red) and HC (blue), with 95% confidence intervals shown in grey.

**Figure 2 ijms-26-10570-f002:**
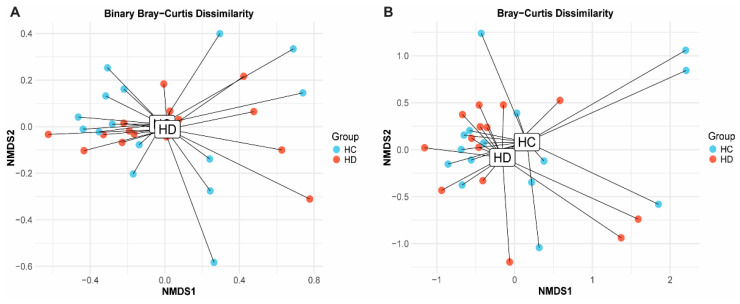
Beta diversity in stool in Hirschsprung’s disease (HD) and healthy controls (HCs). Nonmetric multidimensional scaling (NMDS) plots depict (**A**) Binary Bray–Curtis dissimilarity and (**B**) Bray–Curtis dissimilarity. Children with HD (red) did not differ significantly from HCs (blue) in either measure of beta diversity, as indicated by the comparable diagnostic clusters generated by NMDS plots in (**A**,**B**).

**Figure 3 ijms-26-10570-f003:**
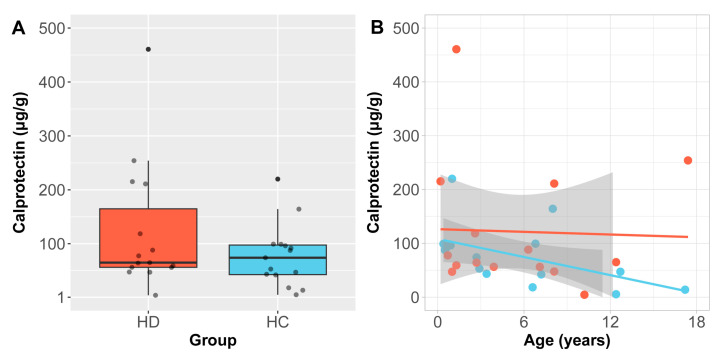
Faecal calprotectin in Hirschsprung’s disease (HD) and healthy controls (HCs). (**A**) Calprotectin did not differ between HD and HC. Boxplot depicts the middle 50% of values, with group medians denoted by horizontal lines. (**B**) Age was negatively correlated with age in the HC group. Solid lines represent generalised linear regression model-predicted values for HD (red) and HC (blue), with 95% confidence intervals shown in grey.

**Figure 4 ijms-26-10570-f004:**
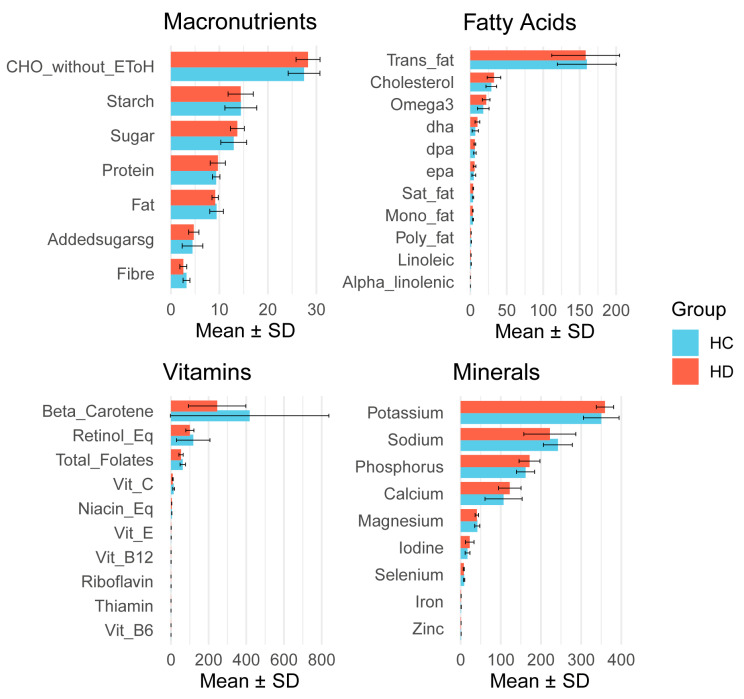
Dietary intake between children with Hirschsprung’s disease (HD) and healthy controls (HCs). Macronutrients are presented as g/1000 kj, fatty acids, vitamins and minerals are presented as mg/1000 kj.

**Table 1 ijms-26-10570-t001:** Demographics of Hirschsprung’s disease (HD) and healthy control (HC) cohorts involved in study.

Demographics	HD (*n* = 15)	HC (*n* = 15)
Age (years)	5.6 (5.0)	5.6 (5.2)
Gender (Male *n*, %) (Female *n*, %)	13 (87%)	13 (87%)
	2 (13%)	2 (13%)
Weight, age-adjusted z score (SD) n.a.	−0.49 (0.9)	0.03 (1.3)
	2	5
Height, age-adjusted z score (SD) n.a.	−0.75 (1.1)	0.07 (1.4)
	3	6
Affected colonic segment length		
Short *n*, (%) Long *n*, (%)	11 (73%) 4 (26%)	
Time since pull-through surgery, yrs (SD) <1 yr 1–4 yrs 5–12 yrs n.a.	4.7 (4.1) 3 5 6 2	
Recent antibiotics * *n*, (%) n.a.	3 (20%) 2	
History of HAEC n.a.	8 (53%) 2	

SD = Standard deviation; HD = Hirschsprung’s disease; HC = healthy control; n.a. = data not available; HAEC = Hirschsprung’s-associated enterocolitis; * Antibiotics administered in the previous 2 months.

**Table 2 ijms-26-10570-t002:** Summary of PedsQL and PedsQL gastrointestinal (GI) symptoms scores separated by age.

Age of Respondent	HD	HC	*p*-Value
PedsQL total score	83 (11.1)	93 (9.2)	0.02
0–12 months	83 (13.9)	85 (11.6)	
13–24 months	87 (2.2)	92 (6.7)	
2–4 yrs	79 (11)	90 (13.6)	
5–7 yrs	79 (8.5)	97 (6)	
8–12 yrs	86 (7.8)	98 (2.4)	
13–18 yrs **	91 (n.a.)	97 (n.a.)	
PedsQL GI symptoms score	83 (10.8)	93 (9.2)	0.005
0–12 months	73 (16)	89 (13)	
13–24 months	89 (10)	91 (7)	
2–4 yrs	83 (9)	90 (14)	
5–7 yrs	80 (7)	97 (6)	
8–12 yrs	86 (8)	98 (2)	
13–18 yrs **	91 (n.a.)	97 (n.a.)	

** Surveys were all parent proxy-reports with the exception of one HD and one HC patient report in the 13–18 yrs age bracket. n.a. only 1 surevey completed for this age bracket so no SD available.

## Data Availability

Data supporting reported results has been deposited in the publicly available NCBI SRA database under BioProject accession number PRJNA1312271.

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
