# Peer review of "Altered Development of Gut Microbiota and Gastrointestinal Inflammation in Children with Post-Operative Hirschsprung’s Disease"

_ijms, 2025, doi:10.3390/ijms262110570_

Round 1
Reviewer 1 Report
Comments and Suggestions for Authors
1.Detailed background information for all samples was not included in the manuscript or supplementary materials.
2.Most samples were collected from males. Does this conform to the clinical onset situation?
3.The age range of the samples is too broad, spanning from a few months to teenagers, with fewer than six samples in each age group. Given that the composition of gut microbiota is strongly correlated with age, have the authors considered this issue?
4. Why was sample HD13 excluded from the HD group? Are there any differences in the background information? Could the large proportion of the Rhodococcus genus from the Actinobacteria phylum in HD13 indicate sequencing contamination?
5. Regarding the analysis of clinical samples, I believe the current sample size is insufficient, with only two female samples in each group. I recommend increasing the number of samples.
Author Response
Thank you for your review. We have addressed all the specific comments but also made some global changes to the discussion to better reflect the results and address the concerns on the studies limitations and implied causality.
Line numbers are for tracked manuscript
Reviewer1
1.Detailed background information for all samples was not included in the manuscript or supplementary materials.
Participant information was included in the methodology section on line 399including inclusion and exclusion criteria for study. Specific cohort information for HD and HC was included in the demographics table (Table 1). A more detailed description of HD cohort has now been added to the results line 134.
“In line with HD epidemiology, more males with HD were recruited than females 13/15 (87%). Of the 15 children recruited, 12 (80%) had short segment length affected and the average number of years (SD) since pull-through surgery was 4.7 (4.1). Five children (33%) had a history of HAEC. The 15 children recruited were age and gender matched with 15 healthy controls (HC). “
2.Most samples were collected from males. Does this conform to the clinical onset situation?
Thank you for highlighting this. This does indeed reflect the HD epidemiology; we have now included this for relevancy in the manuscript on line 135.
“In line with HD epidemiology, more males with HD were recruited than females 13/15 (87%).”
3.The age range of the samples is too broad, spanning from a few months to teenagers, with fewer than six samples in each age group. Given that the composition of gut microbiota is strongly correlated with age, have the authors considered this issue?
Yes, our study design is controlled for age and gender by matching our HD cohort by age and gender to our HC cohort. In our analysis we have also considered age as a confounding variable and included it as a variable in our GLMs. This was outlined in the methods on line 394 and 443. We have changed the wording slightly to improve understanding
“We conducted a cross-sectional, controlled observational study, comparing children with HD to an age and sex-matched cohort of healthy controls (HCs) to minimise demographic confounding.”
“To account for any residual variation not fully addressed by matching, differences between groups were assessed using generalized linear models with age (continuous) included as a covariate.”
We have also considered the low samples numbers as a major limitation of the study and have made this more explicit in the discussion as well as the likely impact on results, line 357.
“Given the high inter-individual variability of the microbiota, the combination of low numbers and high variability may have increased the risk of false negatives due to type II error.”
- Why was sample HD13 excluded from the HD group? Are there any differences in the background information? Could the large proportion of the Rhodococcus genus from the Actinobacteria phylum in HD13 indicate sequencing contamination?
We did not remove from analysis, but due to the limited diversity, we did run sensitivity analysis to investigate if this outlier was influencing the results. As the results did not change with or without the sample included, we left the sample in for subsequent analysis. We have made this clearer in the results section on line 171.
“Removing this sample for leave-one-out sensitivity analysis did not alter the results of statistical comparisons, thus, the sample was not excluded from analysis.”
- Regarding the analysis of clinical samples, I believe the current sample size is insufficient, with only two female samples in each group. I recommend increasing the number of samples.
As a rare disease, the prevalence of this condition is low, inherently limiting recruitment potential. Given the paucity of human studies in this space, we initiated this as a pilot study to determine if there were any broad microbiota changes observed. Whilst we did observe altered microbiota diversity development and inflammatory biomarker trends, we did not find group-wide changes in alpha and beta diversity. Given the inherent microbiota variability between individuals and our low sample numbers, we acknowledge this may be a result of type II errors. We have made this more explicit in the discussion, line 355.
“This study was limited by its small sample size, cross-sectional design, and reliance on stool sampling, which may miss mucosa-associated alterations. The combination of low participant numbers and high inter-individual variability increases the risk of false negatives (type II errors). Nonetheless, careful control for confounding variables including diet, antibiotic exposure, age and gender strengthens confidence in the findings. Importantly, HD is rare and comparable studies in the field have included similar or smaller cohorts, underscoring the challenges of research in this population. All eligible participants meeting the inclusion and exclusion criteria were invited to participate to maximise sample size. ”
Reviewer 2 Report
Comments and Suggestions for Authors
This pilot study addresses a clinically relevant gap by investigating gut microbiota dysbiosis and inflammation in children with post-operative Hirschsprung’s disease. While the hypothesis linking microbial alterations to persistent GI symptoms is compelling, the study is significantly limited by its small sample size, cross-sectional design, and methodological constraints. The findings suggest disrupted age-related microbial development in HD but lack statistical power to confirm core hypotheses. Major revisions are required to address critical limitations before publication. Some specific comments are detailed below.
1. N=15/group is severely underpowered for microbiota studies (typically require >30/group). Negative findings (e.g., no α/β diversity differences, no HAEC-microbiota links) likely reflect Type II error.
2. Abstract/Introduction emphasize HAEC-microbiota links, yet results show no differences in microbiota between HD with/without HAEC history (l. 179-184). This core hypothesis remains unaddressed.
3. Despite documenting dietary intake, diet-microbiota interactions were not analyzed (critical for dysbiosis studies). The suggestion is to include PERMANOVA/ANCOM testing diet (e.g., fiber/fat intake) against microbiota profiles. Discuss diet as potential confounder.
4. Methodological ambiguities - please specify stool homogenization protocol of DNA extraction (critical for reproducibility).
5. Lack diagnostic criteria for HAEC (e.g., HAEC score, histology). Please define "recurrent" (episodes/year?).
6. FDR-adjusted p-values for correlations were missing.
7. Please improve resolution of Fig. 1-3. In Fig. 3B, specify units (µg/g?).
8. Please report p-values for Table 2 and include calprotectin values stratified by age.
9. Please clarify contradictory calprotectin findings: "No HD-HC difference" (Fig. 3A) vs. correlation with Fusobacteria.
10. Discussion section:overinterpretation - please avoid implying causality (e.g., "dysbiosis causesinflammation") without longitudinal/temporal data.
11. Discussion section:the suggestion is to contrast with prior HAEC-microbiota studies (Li et al., Yan et al.) and discuss methodological differences (e.g., stool vs. tissue biopsies, age disparities).
12. Please define all acronyms at first use (e.g., ANCOM, PERMANOVA), check this issue throughout the paper.
13. Please italicize genus/species (e.g., Fusobacteria, Oscillibacter), check this issue throughout the paper.
14. Please distinguish between active HAEC and chronic inflammation (calprotectin elevation ≠ HAEC).
Author Response
Thank you for your review. We have addressed all the specific comments but also made some global changes to the discussion to better reflect the results and address the concerns on the studies limitations and implied causality.
Line numbers are for tracked manuscript
Reviewer 2
This pilot study addresses a clinically relevant gap by investigating gut microbiota dysbiosis and inflammation in children with post-operative Hirschsprung’s disease. While the hypothesis linking microbial alterations to persistent GI symptoms is compelling, the study is significantly limited by its small sample size, cross-sectional design, and methodological constraints. The findings suggest disrupted age-related microbial development in HD but lack statistical power to confirm core hypotheses. Major revisions are required to address critical limitations before publication. Some specific comments are detailed below.
- N=15/group is severely underpowered for microbiota studies (typically require >30/group). Negative findings (e.g., no α/β diversity differences, no HAEC-microbiota links) likely reflect Type II error.
We acknowledge the small sample size and have highlighted the fact that this is a pilot study and that negative findings may be reflecting a type II error in the discussion, line 355. Nevertheless, we would like to reiterate that is a rare disease, and all potential subjects who fulfilled the inclusion and exclusion criteria were approached for recruitment and informed consent to participate in an effort to achieve a larger sample size. Furthermore, previous similar studies in the field recruited smaller number of subjects (Yan et al. (n=4) and Li et al. (n=13)).
“This study was limited by its small sample size, cross-sectional design, and reliance on stool sampling, which may miss mucosa-associated alterations. The combination of low participant numbers and high inter-individual variability increases the risk of false negatives (type II errors). Nonetheless, careful control for confounding variables including diet, antibiotic exposure, age and gender strengthens confidence in the findings. Importantly, HD is rare and comparable studies in the field have included similar or smaller cohorts, underscoring the challenges of research in this population. All eligible participants meeting the inclusion and exclusion criteria were invited to participate to maximise sample size.”
- Abstract/Introduction emphasize HAEC-microbiota links, yet results show no differences in microbiota between HD with/without HAEC history (l. 179-184). This core hypothesis remains unaddressed.
Thank you for highlighting this. We have now addressed this core hypothesis in the abstract. We have also discussed this result and potential limitations in the discussion.
- Despite documenting dietary intake, diet-microbiota interactions were not analyzed (critical for dysbiosis studies). The suggestion is to include PERMANOVA/ANCOM testing diet (e.g., fiber/fat intake) against microbiota profiles. Discuss diet as potential confounder.
We agree that diet is a critical factor in microbiome analysis and therefore included dietary intake of both cohorts in our analysis and checked for dietary-microbiota interactions but did not find anything significant. This was reported in the methods on lines 456, and results on lines 242. We have now moved these results from the supplementary to the main body of the manuscript (Figure 4) to reflect the importance of this analysis, despite the negative findings.
- Methodological ambiguities - please specify stool homogenization protocol of DNA extraction (critical for reproducibility).
We have expanded methodology on line 407 to include homogenisation steps
“All subjects were requested to provide a stool sample, collected as per the EARTH program protocol [18]. Samples were thawed on ice and homogenized for use in DNA extraction using the QIAamp Fast DNA Stool Mini Kit (QIAGEN, Hilden, Germany) as per manufacturer instructions.”
- Lack diagnostic criteria for HAEC (e.g., HAEC score, histology). Please define "recurrent" (episodes/year?).
We have now added this information into methodology on line 435.
“HAEC was defined clinically and based on discharge diagnosis: characteristic symptoms of fever, lethargy, vomiting, and diarrhea, often with blood, and in the absence of infectious cause (negative stool microbiology).”
- FDR-adjusted p-values for correlations were missing.
FDR adjusted p-values were included in the methods (A) line 453 and reported in the results when multiple testing was used in taxa vs calprotectin (B) line 215 and for dietary variables vs taxa (C) line 243. Originally they were reported as “adjusted p”, but have now been changed to q values to signify FDR adjusted p values.
- Differential abundances of taxonomy between cohorts were determined using the ANCOM package v1.1–3 and corrected for multiple testing (false discovery rate (FDR) q<0.05). Correlations between two continuous variables were performed using Spearman correlations (adjusted p-values were produced using the Benjamini-Hochberg correction for multiple testing p<0.05).
- calprotectin was inversely correlated with abundance of the Oscillibacter genus in HCs (Spearman’s rho = -0.84, q = 0.03) and positively correlated with abundance of the Fusobacteria phylum in HD (Spearman’s rho = 0.76, q = 0.02).
- There were also no significant associations between dietary variables and faecal calprotectin (q > 0.42) or between dietary variables and count data for specific phyla or genera (q > 0.07).
- Please improve resolution of Fig. 1-3. In Fig. 3B, specify units (µg/g?).
We have re-done figures to improve readability and accuracy and included an additional figure to show the dietary results. The units are already specified in 3B.
- Please report p-values for Table 2 and include calprotectin values stratified by age.
Table 2 reports the QOL results stratified by survey type as the surveys for each age group are slightly different. As there are limited numbers in each group, p values for each group are not appropriate.
Furthermore, calprotectin results were reported in the results on line 205 and presented across age in figure 3, and is separate from QOL survey results. Hence, we do not think it is appropriate to present the calprotectin data as suggested.
- Please clarify contradictory calprotectin findings: "No HD-HC difference" (Fig. 3A) vs. correlation with Fusobacteria.
Thank you for this. The results you mentioned are not contradictory but illustrate two different findings. The results indicate that while there was no overall significant difference in calprotectin concentrations between the two groups, some specific taxa are either positively or negatively correlated with calprotectin concentrations. We have tried to make this clearer in the text, line 296.
“Although we did not observe broad compositional shifts, exploratory analyses identified taxa of interest. Fusobacteria correlated positively with calprotectin, suggesting a pro-inflammatory role, consistent with reported associations with IBD and pro-inflammatory pathways [18-20]. Oscillibacter showed inverse associations, consistent with protective, anti-inflammatory effects reported elsewhere [21, 22]. While both taxa are relevant to gastrointestinal health, underscoring their potential importance in HD pathophysiology, larger studies are needed to assess causality.
- Discussion section:overinterpretation - please avoid implying causality (e.g., "dysbiosis causes inflammation") without longitudinal/temporal data.
Thank you, we did not mean to imply causality. We have adjusted the discussion to limit overinterpretation.
- Discussion section:the suggestion is to contrast with prior HAEC-microbiota studies (Li et al., Yan et al.) and discuss methodological differences (e.g., stool vs. tissue biopsies, age disparities).
Thank you for highlighting the significance of methodological differences. We agree and already included specific discussion of methodological differences in stool verses tissue and age in the original manuscript and in the limitations paragraph. We have made global changes to the discussion to address concerns from all reviewers, but we still believe we have appropriately addressed methodological differences of other HAEC-microbiota studies in relation to our own (line 329)
“We did not identify stool microbiota signatures predictive of recurrent HAEC, contrasting with tissue-based studies in younger patients at the time of surgery [16, 17]. Differences in biospecimen type [18], timing of sampling, and patient age likely explain these discrepancies. Our single stool collection timepoint was outside of active HAEC, as opposed to stool samples obtained at the time of emergency or routine surgery in Li et al 2016 and Yan Z, et al. Likewise, the average age of participants in our study was 5.6 years (SD 5.0) compared to < 12 months in Li et al 2016 and Yan Z, et al. …..
This study was limited by the small sample size, cross-sectional design, and reliance on stool sampling, which may miss mucosa-associated alterations.”
- Please define all acronyms at first use (e.g., ANCOM, PERMANOVA), check this issue throughout the paper.
All acronyms have been defined.
- Please italicize genus/species (e.g., Fusobacteria, Oscillibacter), check this issue throughout the paper.
All genus/species have been italicized.
- Please distinguish between active HAEC and chronic inflammation (calprotectin elevation ≠ HAEC).
Thank you for highlighting this. We have now added in diagnostic criteria for HAEC into the methodology on line 435 to distinguish between active HAEC and chronic inflammation
“HAEC was defined clinically and based on discharge diagnosis: characteristic symptoms of fever, lethargy, vomiting, and diarrhea, often with blood, and in the absence of infectious cause (negative stool microbiology).”
Reviewer 3 Report
Comments and Suggestions for Authors
Using 16S rRNA gene sequencing and fecal calprotectin measurements, the authors examine microbial diversity, composition, and associations with inflammation, GI symptoms, and diet in 15 children with post-operative Hirschsprung’s disease compared to 15 healthy controls. The study finds that while overall microbial diversity and composition are not significantly altered in HD, typical age-related increases in alpha diversity and decreases in calprotectin are absent, and Fusobacteria abundance correlates with inflammation. The work addresses an important question linking gut microbiota to persistent GI symptoms in HD, but several aspects of study design, analysis, and interpretation limit the strength of the conclusions, indicating the need for revision.
Major comments:
1. The small cohort size (n=15 per group) limits statistical power and generalizability, especially for subgroup analyses such as recurrent HAEC. The authors should discuss this limitation more explicitly and consider emphasizing that findings are preliminary.
2. The cross-sectional design prevents assessment of temporal dynamics in microbiota development. Longitudinal sampling, particularly around the time of surgery and early childhood, would provide stronger evidence for delayed microbial maturation in HD. The manuscript should clearly acknowledge this limitation.
3. The link between microbial diversity, Fusobacteria abundance, and inflammation is intriguing but correlative. The authors should avoid implying causation and consider alternative explanations, including diet, antibiotic exposure, or other unmeasured confounders.
4. The negative findings regarding overall microbial composition and recurrent HAEC may reflect sample type (stool versus tissue) or participant age. The discussion could be strengthened by comparing these methodological differences more directly with previous studies that reported positive associations.
5. Statistical analyses should be more clearly described, particularly the adjustment for multiple comparisons in correlation analyses and ANCOM tests. It would help to include exact p-values and clarify which findings remain significant after correction.
Minor comments:
1. Some figures and tables (e.g., Supplementary Tables 1-3) are referenced but not fully described in the text. Briefly summarizing key findings from these tables would improve readability.
2. Terminology such as “delayed acquisition of microbial diversity” should be tempered to avoid overstatement given the cross-sectional design.
3. Some sentences in the discussion are repetitive, particularly regarding Fusobacteria and Oscillibacter. Condensing these sections would improve flow.
4. Minor typographical issues (e.g., inconsistent spacing, “Rhodoccocus” likely should be “Rhodococcus”) should be corrected.
Author Response
Thank you for your review. We have addressed all the specific comments but also made some global changes to the discussion to better reflect the results and address the concerns on the studies limitations and implied causality.
Line numbers are for tracked manuscript
Reviewer 3
Using 16S rRNA gene sequencing and fecal calprotectin measurements, the authors examine microbial diversity, composition, and associations with inflammation, GI symptoms, and diet in 15 children with post-operative Hirschsprung’s disease compared to 15 healthy controls. The study finds that while overall microbial diversity and composition are not significantly altered in HD, typical age-related increases in alpha diversity and decreases in calprotectin are absent, and Fusobacteria abundance correlates with inflammation. The work addresses an important question linking gut microbiota to persistent GI symptoms in HD, but several aspects of study design, analysis, and interpretation limit the strength of the conclusions, indicating the need for revision.
Major comments:
- The small cohort size (n=15 per group) limits statistical power and generalizability, especially for subgroup analyses such as recurrent HAEC. The authors should discuss this limitation more explicitly and consider emphasizing that findings are preliminary.
As a rare disease, the prevalence of this condition is low, inherently limiting recruitment potential. Given the paucity of human studies in this space, we initiated this as a pilot study to determine if there were any broad microbiota changes observed. We agree that the sample size is very small and may be impacting our ability to find significance in this population. We have highlighted the fact that this is a pilot study and that negative findings may be reflecting a type II error in the discussion (line 355).
“ This study was limited by the small sample size from a single centre. Given the high inter-individual variability of the microbiota, the combination of low numbers and high variability may have increased the risk of false negatives due to type II error.”
- The cross-sectional design prevents assessment of temporal dynamics in microbiota development. Longitudinal sampling, particularly around the time of surgery and early childhood, would provide stronger evidence for delayed microbial maturation in HD. The manuscript should clearly acknowledge this limitation.
We agree that the cross-sectional design prevents assessment of temporal dynamics. However, the spread of samples across the childhood does allow us some ability to evaluate age-related trends between the groups. We have acknowledged the limitation of cross-sectional design, whilst also acknowledging the study design that controlled for age by age and gender matching controls and by the inclusion of age as variable in our GLM models.
- The link between microbial diversity, Fusobacteria abundance, and inflammation is intriguing but correlative. The authors should avoid implying causation and consider alternative explanations, including diet, antibiotic exposure, or other unmeasured confounders.
We have considered diet and antibiotic exposure in our analysis but appreciate there may be many other unmeasured cofounders. It was not our intention to imply causality, so we thank you for highlighting this. We have adjusted the discussion to limit overinterpretation on lines 296 and 355.
“Although we did not observe broad compositional shifts, exploratory analyses identified taxa of interest. Fusobacteria correlated positively with calprotectin, suggesting a pro-inflammatory role, consistent with reported associations with IBD and pro-inflammatory pathways [18-20]. Oscillibacter showed inverse associations, consistent with protective, anti-inflammatory effects reported elsewhere [21, 22]. While both taxa are relevant to gastrointestinal health, underscoring their potential importance in HD pathophysiology, larger studies are needed to assess causality.”
……..This study was limited by the small sample size, cross-sectional design, and reliance on stool sampling, which may miss mucosa-associated alterations. Given the high inter-individual variability of the microbiota, the combination of low sample numbers and high variability may have increased the risk of false negatives due to type II errors. Nevertheless, careful control for confounding variables (diet, antibiotics, age, gender) strengthens confidence in the findings. Future longitudinal studies beginning in infancy, with integrated analyses of microbiota, inflammation, and patient-reported outcomes, are needed to define causal pathways and identify potential therapeutic targets.”
- The negative findings regarding overall microbial composition and recurrent HAEC may reflect sample type (stool versus tissue) or participant age. The discussion could be strengthened by comparing these methodological differences more directly with previous studies that reported positive associations.
Thank you for highlighting the significance of methodological differences. We agree and already included specific discussion of methodological differences in stool verses tissue and age in the original manuscript and in the limitations paragraph. We have made global changes to the discussion to address concerns from all reviewers, but we still believe we have appropriately addressed methodological differences of other HAEC-microbiota studies in relation to our own line 329.
“We did not identify stool microbiota signatures predictive of recurrent HAEC, contrasting with tissue-based studies in younger patients at the time of surgery [16, 17]. Differences in biospecimen type [18] and patient age likely explain these discrepancies with the average age of participants in our study 5.6 years (SD 5.0) compared to < 12 months in Li et al 2016 and Yan Z, et al. Notably, this study is the first to examine microbiota alongside quality of life in HD, highlighting persistent symptom burden despite limited evidence of compositional change…………….
This study was limited by the small sample size, cross-sectional design, and reliance on stool sampling, which may miss mucosa-associated alterations.”
- Statistical analyses should be more clearly described, particularly the adjustment for multiple comparisons in correlation analyses and ANCOM tests. It would help to include exact p-values and clarify which findings remain significant after correction.
FDR adjusted p-values were included in the methods (A) line 453 and reported in the results when multiple testing was used in taxa vs calprotectin (B) line 215 and for dietary variables vs taxa (C) line 242. Originally they were reported as “adjusted p”, but have now been changed to q values to signify FDR adjusted p values.
- Differential abundances of taxonomy between cohorts were determined using the ANCOM package v1.1–3 and corrected for multiple testing (false discovery rate (FDR) q<0.05). Correlations between two continuous variables were performed using Spearman correlations (adjusted p-values were produced using the Benjamini-Hochberg correction for multiple testing p<0.05).
- calprotectin was inversely correlated with abundance of the Oscillibacter genus in HCs (Spearman’s rho = -0.84, q = 0.03) and positively correlated with abundance of the Fusobacteria phylum in HD (Spearman’s rho = 0.76, q = 0.02).
- There were also no significant associations between dietary variables and faecal calprotectin (q > 0.42) or between dietary variables and count data for specific phyla or genera (q > 0.07).
Minor comments:
- Some figures and tables (e.g., Supplementary Tables 1-3) are referenced but not fully described in the text. Briefly summarizing key findings from these tables would improve readability.
We have summarized key findings to improve readability.
- Terminology such as “delayed acquisition of microbial diversity” should be tempered to avoid overstatement given the cross-sectional design.
We have removed this terminology to better reflect the correlation and cross=sectional nature of our results on line 374
“Children with HD diverge form healthy peers in the typical developmental trajectory of gut microbial diversity and inflammation, potentially mediated by Fusobacteria. They also exhibit lower PedsQL and gastrointestinal symptom scores than HC, though these were not directly linked to microbial diversity or inflammation. While these findings are constrained by sample size, they highlight the need for larger, longitudinal and mechanistic studies integrating microbiota, inflammatory, and clinical outcomes to define causal pathways and identify therapeutic strategies to improve gastrointestinal health and quality of life in children with HD.”
- Some sentences in the discussion are repetitive, particularly regarding Fusobacteria and Oscillibacter. Condensing these sections would improve flow.
Thank you we agree, we have now refined and condensed this section.
“Although we did not observe broad compositional shifts, exploratory analyses identified taxa of interest. Fusobacteria correlated positively with calprotectin, suggesting a pro-inflammatory role, consistent with reported associations with IBD and pro-inflammatory pathways [18-20]. Oscillibacter showed inverse associations, consistent with protective, anti-inflammatory effects reported elsewhere [21, 22]. While both taxa are relevant to gastrointestinal health, underscoring their potential importance in HD pathophysiology, larger studies are needed to assess causality.”
- Minor typographical issues (e.g., inconsistent spacing, “Rhodoccocus” likely should be “Rhodococcus”) should be corrected.
Thank for drawing our attention to these errors. They have been rectified throughout the manuscript.
Round 2
Reviewer 1 Report
Comments and Suggestions for Authors
There are no suggestions or comments.
Reviewer 2 Report
Comments and Suggestions for Authors
The authors have greatly improved the manuscript according to my previous comments and suggestions. There is no other comment for the moment, the current version is accepted for publication.
Reviewer 3 Report
Comments and Suggestions for Authors
The authors have provided clear, thorough, and well-considered responses to all previous comments. The revised manuscript demonstrates significant improvement in clarity, methodological transparency, and interpretation. Key issues related to study design, statistical analysis, and potential overinterpretation have been appropriately addressed. The discussion now provides a balanced and nuanced assessment of the findings, with careful acknowledgment of limitations and alignment with existing literature. Overall, the authors have been highly responsive, and the manuscript now meets the standards for publication.